# Development of Landscape Architecture Design Students' Pro-Environmental Awareness by Project-Based Learning

**Juan Xi [1] and Xinjun Wang [2],***

[1] School of Foreign Studies, Changzhou University, Changzhou 213159, China; 00002043@cczu.edu.cn
[2] School of Art and Design, Changzhou Institute of Technology, Changzhou 213022, China
* Correspondence: wxinjun@cit.edu.cn

**Abstract:** With a rapid pace of urbanization, urban environment problems have affected large numbers of people and aroused widespread concern. Landscape architecture design helps improve the welfare of urban residents and ecological function of urban green spaces. Course-based influence of college students majoring in landscape architecture is an efficient way to prepare future landscape architects with environmental awareness. This research applied project-based learning (PBL) and zone of proximal development (ZPD) in the landscape architecture design course, and developed the investigation–design–construction PBL modules of the course. The experimental group, 57 college juniors majoring in landscape architecture, received PBL education while learning the course. At the end of the course, a questionnaire was answered by the junior students and the control group, which comprised 60 senior students who received no PBL instruction while learning the same course in the third year. The results indicate that the PBL approach was well accepted by 90% of the experimental group, who were 17.37% more likely to employ pro-environmental design methods in their future work than the control group. It was also found that employing the PBL approach in the landscape architecture design course had a positive influence upon students' pro-environmental values, knowledge and attitudes.

**Keywords:** PBL; landscape architecture design; pro-environmental awareness; environmentally unfriendly behavior

## 1. Introduction

With rapid urbanization, the urban population has risen dramatically around the world, from 13% of the whole population in 1900 to 66% of the whole population by 2050 [1]. The concentration of population in cities causes massive energy consumption and carbon emissions, therefore, building greener cities is vital to accomplishing the goal of carbon neutrality and alleviating global climate crises. Urban parks are a valuable resource in building sustainable cities, as they offer environmental, economic and social advantages [2]. Moreover, they provide pleasant spaces for physical exercise and contribute to promoting city residents' health and reducing risks of developing chronic diseases [3,4]. Landscape architecture students are the main force of future urban landscape design and construction, and it is of considerable educational, environmental and economic significance to develop their pro-environmental awareness.

Pro-environmental awareness is the awareness of environmental issues and measures to be taken to bring about good practices towards environment conservation [5]. It is the main driving factor of pro-environmental behaviors [6]. Pro-environmental awareness is composed of pro-environmental knowledge, values and attitudes [7,8]. Pro-environmental values positively influence pro-environmental attitudes; pro-environmental attitudes and knowledge positively influence pro-environmental behaviors, and meanwhile, pro-environmental knowledge also influences pro-environmental values and attitudes [7]. A lack of education contributes to public apathy to climate problems. Higher education

plays a key role in enhancing students' pro-environmental awareness [7,9]. Sufficient pro-environmental knowledge and skills are essential to sustainable human development in the future. However, pro-environmental awareness-related goals are not easily achieved through the traditional mode of design education where teachers are "the transmitter of the knowledge" while students act as "the receptor of the information" [10], as a result of which, students, unable to fully engage in the learning process, can only have a superficial understanding of the knowledge and skills they need to command [11].

More institutions are ready to shape a sustainable society through education, but there has been a limited number of them providing sustainability education curriculums in the past two decades [12]. Designers are faced with increasingly complex and impactful challenges, however, the current design education does not always prepare them for these challenges [12,13]. An online survey of Metropolis magazine readers that are mostly architects, interior designers and a few landscape architecture designers, revealed a strong interest in sustainable design (93%), but a lack of education and training caused 70% of the survey respondents to feel not qualified to take a job where sustainable design is needed [14]. A study of 100 interior designers by the International Interior Design Association found that only 37% of their projects contain sustainable solutions. The obstacles of their design practices were "too little information" and a lack of research demonstrating the economic benefit of sustainable design [15]. However, if the landscape architecture designers do not understand the relationship between landscape elements and their environmental effects, their designs may create inadvertent modifications that can even make the situation worse [16]. Training landscape architects with adequate knowledge and the skills of pro-environmental design has thus gained more prominence and urgency. The challenges in this era make it necessary to "move away from a nineteenth-century model of architectural education to one that is relevant for today" and "to tackle the major problems that society currently faces", among which climate is one wanting immediate attention [17].

Given the global attention on climate change, different educational approaches integrating sustainability in architectural programs have been implemented around the world [18–20]. Successful strategies include integrating sustainability in already existing courses, improving students' sustainability awareness [7] and creating sustainability-specific courses [16,21]. For instance, climate-responsive landscape architecture design classes were introduced in Wageningen University and University of Guelph, and practice-oriented teaching and learning methods were used to encourage students to accumulate climate knowledge, analyze climate-related problems in a study site and finally generate climate-appropriate design solutions [16]. With the whole international community exhibiting growing concern over climate issues, environmentally friendly design will assume considerable significance in the work of landscape architecture professionals. Students majoring in this area of study, as the talent reserve for the industry, will find it essential, presumably in an increasing number of occasions, to take environment and climate into account in their future work of urban planning and green space design. Enhancing these to-be professional designers' pro-environmental awareness is conducive to the creation of a more ecologically livable urban environment. The learning-by-doing paradigm for design education was widely used in architecture design studios [22], but reflection was seldom introduced [23]. Given the practicality and complexity of landscape architecture design, the traditional lecture-based mode of teaching and learning has to give way to open-ended process-oriented approaches that emphasize student needs, inspire active learning and relate school learning to present-day problems. This is where project-based learning (PBL) can come in.

PBL has its theoretical foundation in constructivism. It has a more positive impact on students' academic achievement than the traditional mode of instruction, but only 20% of studies on its application have been conducted in higher education [11]. Some research suggests that it is an innovative teaching and learning strategy that can improve student learning in higher education [11,24]. PBL is a recommendable educational approach for the development of comprehensive learner competences, linking teaching with the

professional sphere and offering multiple chances for students to develop their technical, contextual and behavioral competences [25]. Learners explore problems in groups and learn to apply and organize knowledge through data collection and discussion to develop their understanding of the knowledge and their ability to apply the knowledge to practical work [26]. PBL is an effective pedagogy in engineering education, as most engineering jobs entail design and practice [27]. PBL has six features, including driving questions, learning goals, participation in education activities, collaboration among students, the use of scaffolding technologies and the creation of tangible artifacts [11], among which the last one is crucial and distinguishes PBL from other education pedagogies [28]. The creation process requires learners to work together and solve problems in the process of knowledge integration, application and construction [11].

When it comes to the specific design of PBL teaching and learning, scaffolding strategy can be drawn from the theory of zone of proximal development (ZPD), which, developed by Russian psychologist Vygotsky, is a widely used educational strategy primarily applied to collaborative learning [29,30]. Vygotsky defined ZPD as the distance between the actual developmental level as determined by independent problem solving and the level of potential development as determined through problem solving under adult guidance or in collaboration with peers. Full development of the ZPD depends upon full social interaction, and the cognitive ability that can be developed with teacher guidance or peer collaboration exceeds what can be achieved alone [31]. Scaffolding is important in maximizing student learning [31]; the purpose of scaffolding strategies in PBL is to support students' content learning and project progress [32]. Some PBL course frameworks were combined with the ZPD [24,32], for instance, a learning community was set up, students and instructor included, to work out a solution to the proposed question or problem [24].

This research integrates PBL and ZPD in landscape architecture design, a compulsory course for landscape architecture undergraduates. The aim is to propose a systematic way to apply PBL at the curricular level and find out if it facilitates students' environmental awareness development.

## 2. Materials and Methods

### 2.1. Participants

Participants of this research comprised college juniors and seniors majoring in landscape architecture design in the Changzhou Institute of Technology, Jiangsu Province, China. The experimental group was 57 juniors who received project-based learning of pro-environmental awareness and design skills in their learning of a compulsory major course, landscape architecture design, before completing a survey questionnaire on their pro-environmental awareness. The control group was 60 seniors who did not receive pro-environmental project-based learning when they learned the same course in their third year at college, but who were invited to complete the same questionnaire. The course lasted 16 weeks, with 4 lessons scheduled every week. The students had learned some prerequisite courses before, such as computer aid design, landscape botany and sketch and basic design.

### 2.2. Curricular Structure

There are 64 lessons in landscape architecture design, 18 of which are theoretical lessons arranged for teaching and learning landscape architecture design theories and methods and 6 of which are evaluations for students' performance after each project phase. The other 40 lessons are arranged for practice activities, and in this study, students participated in a landscape architecture project.

To enhance students' pro-environmental awareness, newly proposed concepts related to pro-environmental landscape design were integrated into the teaching content, and taught in theoretical modules of the course. Project tasks were arranged in the PBL modules of the course, including investigation of public parks in the city of Changzhou and building a garden on the college campus following the pro-environmental design. The content in

the theoretical modules, including advantages of urban greenspaces and design methods and cases, laid the foundation of activities in PBL modules. In the PBL modules, students discovered and analyzed problems from local urban parks and combined solutions in their work of campus garden design and construction. The application of the PBL approach in the PBL modules was intended to give the students opportunities to put theoretical knowledge into practical use and to help internalize pro-environmental awareness in them.

The evaluation is the analysis and internalization process of the learning at the end of each phrase of the project. This study not only summarized and analyzed the students' performance in each phase of the project but also compared the results in the whole class. The students' pro-environmental awareness was evaluated by questionnaire at the end of the course. The content and structure of the project in landscape architecture design are as shown in Figure 1.

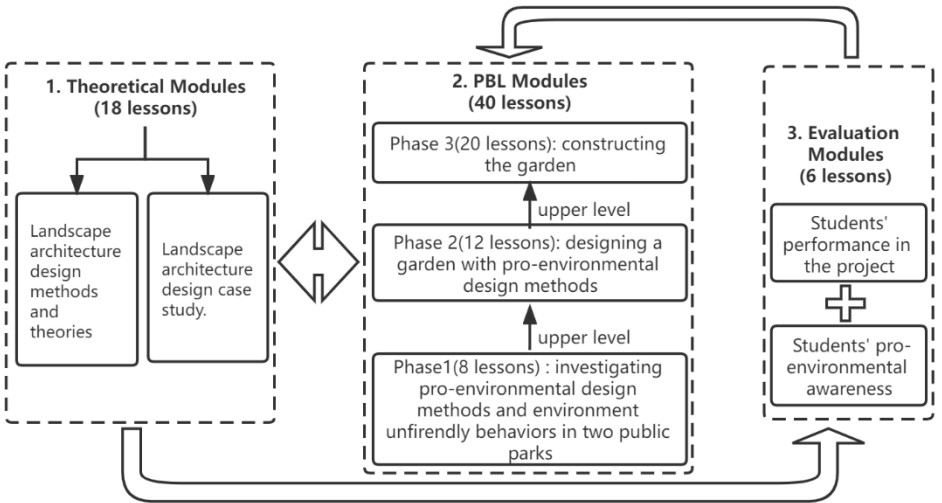

**Figure 1.** The content and structure of landscape architecture design.

### 2.3. Content Design of the Project Modules

The content of the course was designed under guidance of PBL and ZPD. Based on the propositions of the pedagogic theories, it is of necessity to integrate appropriately designed project modules in the teaching and learning content of a course to gradually expand the upper limit of students' proximal development zone. The number of project modules included in a course depends on the teaching objectives, class hours, students' academic status quo and the level of project difficulty. In the PBL modules of the landscape architecture design course, three closely linked learning phases were designed to gradually deepen the students' understanding of pro-environmental design skills, to develop their pro-environmental awareness step by step and to continuously push the upper limit of their learning ability (Figure 1).

Phase One lasted two weeks (8 lessons). It was a survey of two parks, Xinqu Park and Zijing Park of Changzhou, Jiangsu Province, China in March and April, 2021. Both parks are public urban greenspaces located in the central area of the city. Xinqu Park is 8.64 hectors in area, and Zijing Park is 25.02 hectors. There were two objectives for assigning students this survey task. One was to help them better understand pro-environmental and environment-unfriendly behaviors of visitors. Pro-environmental behaviors are behaviors that an individual does to reduce environmental destruction and to conserve the environment [5], and environmentally unfriendly behaviors are the behaviors detrimental to the environment. In the survey task, students discovered environmentally unfriendly behaviors visitors performed in the parks.

The incidence of environmentally unfriendly behaviors was used as an indicator because of different areas of the parks. In Equation (1), *I* means the incidence of environ-

mentally unfriendly behaviors found in the parks, *Ai* refers to the behaviors found in parks and *s* is the area of the investigated parks.

$$I = (\sum_{k=1}^{n} Ai)/s \tag{1}$$

The other objective was to help them investigate the pro-environmental design schemes in the parks. The 57 junior students were divided into 11 groups, with 5 or 6 in a group, led by the group leader elected by the students themselves. The investigation data were analyzed by SPSS 23.

Phase Two of the project lasted three weeks (12 lessons). It was to design a small green space on the college campus. The students were required to integrate in their garden design the environment-related knowledge that they had gained from the park investigation. The design site is a flat plot of land of about 200 square meters in the Changzhou Institute of Technology. Students needed to consider creativity, function(s), applicability and standardization of their designs, and meanwhile consider whether their designs could get across the pro-environmental design philosophy to others.

Phase Three lasted five weeks (20 lessons), during which students constructed the garden they had designed. The tasks needed to be completed before the end of the course. Firstly, the landscape architecture design plan was selected by the course teacher and external tutors. Then, the construction work was arranged based on the process of the real landscape architecture project, so students could face real problems that they would meet in future design practices, such as material selection, project cost, project schedule, etc. The difference from the real project was that in the course the students were required to use the pro-environmental design method in their landscape architecture design. The three phases of the project were tightly connected. Phase 1 allowed students to fully understand the relationship between landscape design and pro-environmental behaviors, which expanded their upper level of the knowledge about pro-environmental values and pro-environment design methods. Phase 2 required students to combine the knowledge they had gained from Phase 1 with their design plans, which also expanded the upper level of their pro-environmental design knowledge and skills. The construction phase gave students an opportunity to fill the gap between landscape architecture design and construction. The landscape architecture design course trains students for future professional work. Engaging the students in a project enhances their understanding of the profession by collecting data and incorporating the results into their design and construction. It is a powerful learning experience for the students and also advances the profession as a whole.

### 2.4. Evaluation and Questionnaire

The evaluation modules took 6 lessons and was composed of two parts. One was students' performance in different phases of the project, and the other was pro-environmental awareness evaluation of the experimental and control groups. External tutors were invited from a local landscape architecture company to summarize and analyze the students' performance in three phases of the project module, together with the teachers and students of the course. The evaluation module was important because comparison between students and timely feedback from tutors and peers would encourage their performance in subsequent project phases, and in the meantime, discussion between students and tutors could enhance understanding of professional knowledge in the process.

To find out if the PBL approach used in landscape architecture design helped enhance students' pro-environmental awareness, the teachers of the course designed a questionnaire to evaluate students' pro-environmental awareness. The 57 junior students, i.e., the experimental group, responded to the questionnaire. Sixty senior students majoring in the same field of study responded to the questionnaire as well. These students had also taken the same course in their third year of undergraduate study, but had not received any PBL instruction with regard to pro-environmental landscape architecture design.



Students' learning outcomes were surveyed through a questionnaire. Both Likert scales from 1 (very low) to 5 (very high) [33] and qualitative open-ended questions were adopted [34]. To measure the junior students' acceptance of the PBL approach, 5 question items were developed (adaptability, methods, emotions) (Supplementary Materials, Q1–Q5). Five question items were developed to measure students' pro-environmental values and knowledge (biosphere, egocentric and anthropocentric) [35]. Pro-environmental attitudes were measured by 2 question items (Supplementary Materials, Q6–Q12).

Cronbach's alpha was used to test the reliability of the questionnaire. It has been recommended that for purposes of group comparisons Cronbach's alpha should reach 0.70 or larger [36,37]. Cronbach's alpha is minimally acceptable between 0.65 and 0.70, acceptable between 0.70 and 0.80 and stronger between 0.8 and 0.9 [7]. The Pearson chi-squared statistical test is a method that determines the significant difference between the expected values and the observed values, and there is dependence between the variables when the *p*-value is less than or equal to the significance level [38,39]. All data were analyzed by SPSS 23.

## 3. Results

### 3.1. Environmentally Unfriendly Behaviors in the Parks

Through the investigation, three types of environmentally unfriendly behaviors were found in the parks: vegetation damage, public facility damage and environment quality damage, as seen in Table 1.

**Table 1.** Environmentally unfriendly behaviors in the investigated parks.

| | Types of Environment-Unfriendly Behavior in Parks | Specific Behaviors | Zijing Park | $I_1$ (per 100 m$^2$) | Xinqu Park | $I_2$ (per 100 m$^2$) |
|---|---|---|---|---|---|---|
| 1 | Vegetation damage | Breaking branches and defloration | 52 | 2.08 | 33 | 3.82 |
| | | Trampling the lawn | 37 | 1.48 | 25 | 2.89 |
| | | Scribbling on the trunk | 7 | 0.28 | 5 | 0.58 |
| 2 | Public facility damage | Damaging roads | 12 | 0.48 | 7 | 0.81 |
| | | Damaging leisure facilities (seats, gallery, etc.) | 3 | 0.12 | 2 | 0.23 |
| | | Damaging guardrails | 2 | 0.08 | 1 | 0.12 |
| | | Damaging billboards | 3 | 0.12 | 5 | 0.58 |
| 3 | Environment quality damage | Spitting | 38 | 1.52 | 33 | 3.82 |
| | | Littering | 65 | 2.60 | 42 | 4.86 |
| | | Posting ads | 0 | 0.00 | 2 | 0.23 |
| | | Graffiti | 5 | 0.20 | 3 | 0.35 |
| | | Excessive noise | 6 | 0.24 | 2 | 0.23 |

$I_1$ is the incidence of Zijing Park. $I_2$ is the incidence of Xinbei Park.

Environmentally unfriendly behaviors were found mainly in the destruction of park plants and environment quality, with 96 (3.84 per 100 m$^2$) and 63 (7.29 per 100 m$^2$) plant-related unfriendly behaviors, and 114 (4.56 per 100 m$^2$) and 82 (9.49 per 100 m$^2$) environment-quality-related unfriendly behaviors found respectively in Zijing Park and Xinqu Park. Fewer public-facility-related unfriendly behaviors were found in the parks. Flowers were more likely to be plucked where flowering plants were close to roads, while plants behind rails or planted farther away from roads were less likely to be damaged. In public parks, people are not allowed to tread on lawns as trampling affects the growth of the grass. Yet in the investigation, students found children run, chase and frolic on the park lawns with no prominent notice boards, while the lawns with such signs suffered less trampling.

Littering and spitting were frequent environment-quality-related behaviors in the parks. A lot of garbage, such as food packaging, was spotted near the small grocery shops in the parks. More spitting behaviors occurred in the smoking and resting areas of the

parks. Placing more dustbins in these areas as well as strengthening management and publicity can alleviate this problem. Xinqu Park saw a greater incidence of environmentally unfriendly behaviors because it is smaller than Zijing Park and surrounded by dense residential buildings, and thus receives a larger flow of visitors.

### 3.2. Pro-Environmental Design Methods Used in the Parks

Students also analyzed pro-environmental landscape design methods used in the two parks. Employment of these methods in urban green space design can visibly protect and improve the surrounding environment. Such methods include using recyclable, or locally produced materials, planting native plants, placing rails and notice boards, etc., as shown in Table 2. The investigation could give students an in-depth understanding of the environmental function of urban greenspace design, and inspire them to use these design methods in their future landscape architecture design practices.

**Table 2.** Pro-environmental methods used in the parks.

| | Pro-Environmental Design Methods Used in the Parks | Application | Function(s) |
|---|---|---|---|
| 1 | Using recyclable materials | Precipitation collection such as permeable pavement and ponds. Recyclable paving materials, such as plastic wood flooring | Saving water and energy Mitigating urban heat island effect Saving materials |
| 2 | Using locally produced materials | Stones for paving Planting native trees, grass and flowering plants | Reducing energy consumption and exhaust emissions in transportation Low cost and easy to maintain |
| 3 | Using low-cost materials | Using wild plants where appropriate Ecological revetment | Easy to maintain Saving materials |
| 4 | Using facilities | Notice boards and rails | Discouraging environment-unfriendly behaviors |

In the investigation, the students developed a deeper understanding of environmentally unfriendly behaviors in urban greenspaces and realized the importance of pro-environmental landscape architecture design methods. The investigation in Phase 1 of the project laid the foundation for the following teaching and learning activities.

### 3.3. The Pro-Environmental Landscape Archhitecture Design

In Phase 2 of the PBL modules, the students were asked to use pro–environmental design methods to design a small garden for an open space of about 200 square meters on the college campus. The students' investigation in Phase 1 of the project brought them more knowledge and inspiration about the pro-environmental landscape architecture design. Some groups used recyclable materials to design exhibition and learning spaces on the plot of land some used local materials and native plants and some designed a pond to collect precipitation. Due to the limited area of the site, not all pro-environmental methods the students had learned from the investigation phase could be used in the design of the campus greenspace. The final design plan selected the most feasible and environment-friendly design schemes produced by the students.

### 3.4. The Construction Activities

The implementation of the design plan enabled students to learn more professional knowledge. It was also a tentative attempt for the teachers. The student groups had to pull together and finish their share of the construction work. Professional construction workers from a local landscape architecture company would offer help when the students found

some construction work too professional for them to complete, such as building the scenery walls of the garden.

The students first set the lines on the site according to the final design plan, and then decided on the foundation heights of the scenery walls and stone walls. Afterwards, they excavated and filled the foundation of the scenery walls with concrete to ensure stability. The most difficult part was to build the scenery walls, where professional bricklayers cooperated with the students to complete the task, and the students also learned a lot of masonry skills from the professional workers during this process. After building the scenery walls, the students levelled the land and applied fertilizers. The last step was sowing flower seeds and planting vegetation, as shown in Figure 2. Different tasks were assigned to every group, and the group leader coordinated and organized the members to complete the construction work on time without affecting the schedule of the next group. The group leaders also communicated and coordinated with one another and prepared the tools and materials needed to complete the tasks in advance. The construction of the park lasted for nearly 2 months, during which the students' design works turned from paper to reality, and the students gained joy as well as professional training.

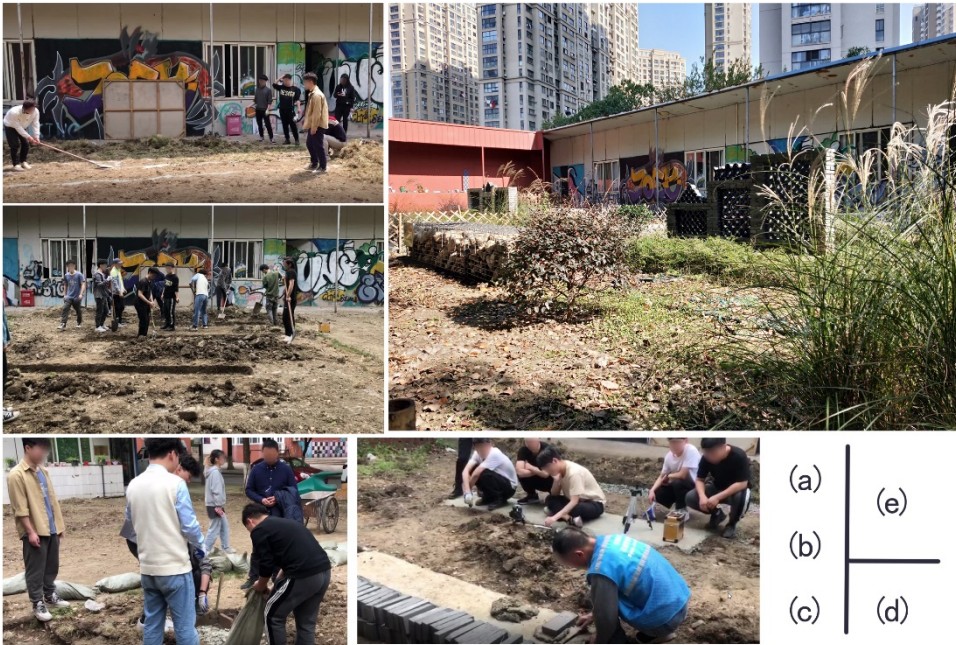

**Figure 2.** Garden construction scenes: (**a**) setting the design on the site; (**b**) digging the foundation of walls; (**c**) filling concrete in the foundation; (**d**) learning bricklaying from external tutors; (**e**) the completed garden. To conserve energy and protect the environment, students used local materials such as stones, plants, bricks and organic mulch, which were easy to procure, transport and maintain. (Images are reproduced with the permission from Xinqian Liu, a student of the experimental group who took these photos.)

*3.5. Students' Attitudes toward the PBL Approach*

Junior students' attitudes toward PBL were evaluated through $Q_1$–$Q_5$ of the Questionnaire ($Q_1$–$Q_5$, $\alpha = 0.876$, Table 3). A total of 65% of them strongly agreed and 26% of them agreed that the PBL method was feasible ($Q_1$). The PBL modules left a deep impression on all the junior students, and almost all of them thought that the grading method was reasonable ($Q_2$, $Q_3$). In total, 93.00% of them believed that external tutors gave them more tips about future careers through communication in the course ($Q_4$). When asked if the PBL approach was recommendable for the learning of other courses, 64.91% of them chose "Strongly Agree", 26.23% chose "Agree" and 8.77% chose "Indifferent" ($Q_5$), as shown in Figure 3, Supplementary Materials.

**Table 3.** Construct reliability of scaled variables.

| Questionnaire | Variables | Question Items | Cronbach's $\alpha$ | | Chi-Squared Test for Significance | |
|---|---|---|---|---|---|---|
| | | | Junior | Senior | $\chi^2$ | *p*-Value |
| 1,<br>Project-based<br>teaching methods<br>($\alpha$ = 0.876) | Adaptability | $Q_1$, The PBL approach is feasible. | 0.769 | | | |
| | | $Q_2$, The project has left a deep impression on me. | 0.876 | | | |
| | Methods | $Q_3$, The evaluation of my performance in the course is reasonable. | 0.792 | | | |
| | | $Q_4$, Communication with external tutors deepens my understanding of the industry. | 0.888 | | | |
| | Emotions | $Q_5$, The PBL approach is recommendable for the learning of other courses. | 0.808 | | | |
| 2,<br>Awareness of<br>pro-environmental design<br>($\alpha_{junior}$ = 0.849<br>$\alpha_{senior}$ = 0.925) | Values and knowledge | $Q_6$, Pro-environmental design is important to conserve urban environments. | 0.780 | 0.886 | | |
| | | $Q_7$, Pro-environmental design helps to reduce the emission of carbon dioxide. | 0.826 | 0.895 | 13.775 ** | 0.003 |
| | | $Q_8$, Pro-environmental design helps to improve the welfare of city residents. | 0.803 | 0.899 | | |
| | | $Q_9$, I will use local materials in my landscape design. | 0.816 | 0.901 | | |
| | | $Q_{10}$, I will use precipitation collection methods in design to save water resources. | 0.801 | 0.885 | | |
| | Attitudes | $Q_{11}$, I will use pro-environmental design in my future career. | 0.827 | 0.932 | 8.198 * | 0.042 |
| | | $Q_{12}$, Pro-environmental design is important for college students majoring in landscape architecture design. | 0.892 | 0.911 | 11.900 ** | 0.008 |

* $p < 0.05$, ** $p < 0.01$.

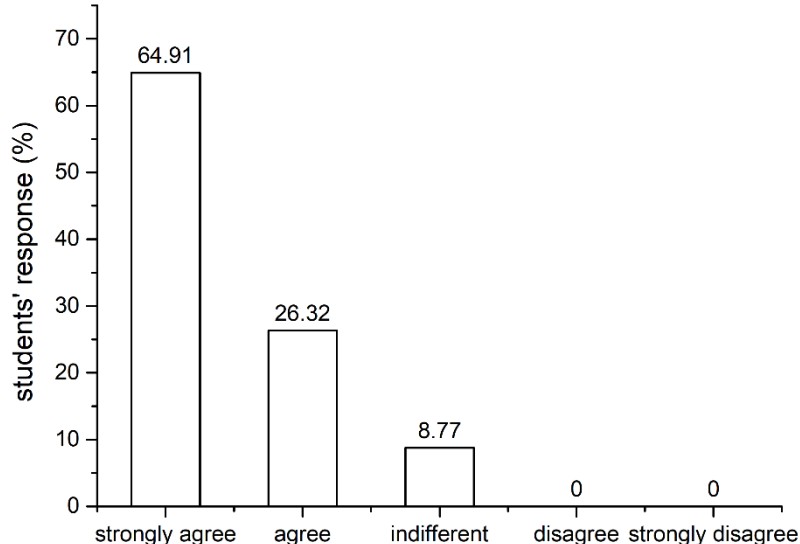

**Figure 3.** The experimental group's responses to $Q_5$.

### 3.6. Students' Attitudes toward Pro-Environmental Design Methods

$Q_6$–$Q_{12}$ of the questionnaire were used to learn about students' awareness toward pro-environmental design methods (Table 3, Supplementary Materials). The reliability was tested by Cronbach's alpha ($\alpha_{junior}$ = 0.849, $\alpha_{senior}$ = 0.925). In the significance test of the control group and experimental group, *p*-values were less than the specified significance level ($Q_6$–$Q_{10}$ and $Q_{12}$, $p < 0.01$; $Q_{11}$, $p < 0.05$). On the value of pro-environmental design, the percentage of students choosing "Indifferent" and "Disagree" in the control group is obviously higher than that of the experimental group. $Q_6$ to $Q_8$ of the questionnaire were about the value of pro-environmental design, which received 13.90% "Indifferent" and 3% "Disagree" in the control group but just 4.1% "Indifferent" in the experimental group. A comparison of the responses from the experimental group and control group showed that the experimental group's understanding of pro-environmental design was enhanced with PBL learning of the course. $Q_9$ and $Q_{10}$ were used to find out the student's pro-environmental knowledge. In total, 7.5% of the control group chose "Indifferent",

while none of the experimental group chose "Indifferent". Instead, all the students of the experimental group chose "Strongly Agree" or "Agree". Compared with the control group, students in the experimental group had learned more knowledge about pro-environmental design through PBL education in the course. $Q_{11}$ and $Q_{12}$ were designed to learn about the students' attitudes toward using pro-environmental design methods in their career. 80.70% of the experiment group and 63.33% of the control group strongly agreed to use pro-environmental design methods. 85.97% of the experimental group strongly agreed that the pro-environmental design methods were important for college students majoring in landscape architecture design, while 66.67% of the control group strongly agreed to this item.

In this research, it is interesting to note that students that had PBL pro-environmental education in the course were more likely to approve of the pro-environmental landscape design methods than students that did not have such education. In terms of the attitudes toward the importance of learning the design methods ($Q_{12}$, $p < 0.01$), the percentage of the experimental group choosing "Strongly Agree" was higher by 19.30% than the control group, yet the percentage of the experimental group choosing "Agree" was almost 1% lower than the control group. No student in the experimental group chose "Indifferent" while 13.33% of the control group chose it, and 5% of the control group chose "Disagree", as shown in Figure 4. It can be inferred that college students majoring in landscape architecture design already have a degree of pro-environmental awareness, and that PBL pro-environmental education in the course can significantly improve their pro-environmental awareness. Pro-environmental awareness influences pro-environmental behaviors [7]. The PBL approach, systematically employed to the course, can galvanize students into pro-environmental design action in their future work.

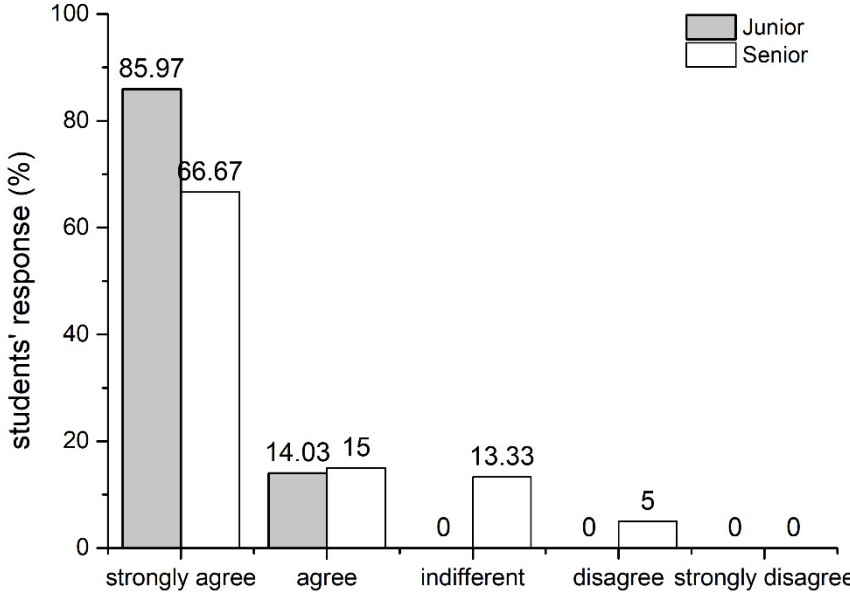

**Figure 4.** The experimental and control groups' responses to Q12.

## 4. Discussion and Conclusions

Students who major in landscape architecture design are key stakeholders in sustainable development, and more efforts are needed to raise their awareness to protect the essence of sustainable development [40–42]. Previous research has proposed three steps to teach pro-environmental design methods in the landscape architecture design course: firstly, to help students accumulate and summarize climatic knowledge about the building site for a green space, secondly, to teach students to analyze the site and identify possible climate-related problems, and finally, to guide students to generate environment-friendly design solutions [16]. Compared with such an approach, the PBL approach used in this

research also emphasized the importance of knowledge accumulation, problem analysis and sequential learning, yet it does not end with generating environment-friendly design solutions but goes further and encourages students to apply their solutions to real construction work and create a tangible landscape architecture artifact. Learning the landscape architecture design course by integrating PBL and ZPD enables students to improve their comprehensive ability, and very importantly, enhance their pro-environmental awareness.

In this research, the project modules of investigation–design–construction were developed following the guidance of PBL and ZPD in the landscape architecture design curriculum. The PBL modules provided students with opportunities to identify environmentally unfriendly behaviors occurring in parks, to produce pro-environmental design schemes that helped discourage such unfriendly behaviors and to review and improve their design schemes in real construction work. This immersive learning experience was well accepted by students and laid a meaningful foundation for their future design practices. Students completed project tasks in groups and interacted regularly so that they could help one another review their design schemes with a fresh pair of eyes [17]. Personal responsibility and satisfaction derived from group success increased students' motivation to learn. External tutors encouraged students' reflection on learning in the implementation of the project and the evaluation of their learning. Students acquired pro-environmental knowledge and developed positive pro-environmental values and attitudes in the project. It is, therefore, safe to conclude that systematic application of PBL in landscape architecture design helps expand students' learning potential and improves their pro-environmental awareness.

Landscape architecture design is a complex and interdisciplinary profession [43]. The education of landscape architects should be based on the knowledge of natural sciences, including landscape ecology, landscape techniques, construction, botany, etc. [44]. In order to better understand the pro-environmental design methods, students need to be exposed to many aspects of guidance, which should not only come from teachers in the classroom, but also can be accessed by bringing in experts from the construction industry [17]. These external tutors are an added value to landscape architecture design education in light of the professional experience that they can transfer to the students, though they may not share the same understanding and knowledge about pro-environmental design methods [21]. Tutors from landscape architecture companies can provide pro-environmental design methods and cases and remind students of many details in project implementation from the perspective of on-site construction. For instance, in Phase Three of the project, the junior students learned the process of building scenery walls from the master bricklayers invited from a local landscape architecture company. First, the bricks must be soaked in water to increase the humidity and make them tightly bond with the concrete. After that, the bricks should be staggered in the long direction, using thin lines as contour lines to ensure that they will not be skewed.

Development of landscape architecture students' pro-environmental awareness in higher education will make more environment-friendly design practices possible in the future. This research shows that the proportion of students that strongly approved of using pro-environmental design methods in the future increased by 17.37% after completing the project in the course. Though it was the first time that the PBL approach had been used in the landscape architecture design course at the Changzhou Institute of Technology, the junior students' responses to the questionnaire show that the approach was well accepted and recognized by more than 90% of them. In terms of the importance of landscape architecture design to climate change, 70% of the surveyed students considered it very important [21]. The percentage of students who recognized the significance of landscape architecture design for environment was 85.97%.

Lack of training and education is one of the obstacles for sustainable landscape architecture design [45]. The PBL approach can be used to enhance landscape architecture undergraduates' pro-environmental awareness. It is also a feasible way to train professional landscape architects and make them more aware of the meaning of pro-environmental

design for helping address environment issues that now concern city administrators and residents.

The investigation–design–construction modules of the project used in this pedagogic research might also be used by teachers of landscape architecture design from other colleges, except that constructing a small garden is demanding in that lots of materials such as concrete, bricks, stones, vegetations and all kinds of tools are needed. Yet, it is possible to take advantage of accessible local resources to build a simple one. It is best to schedule the construction work in warm months so that more options of flowering plants are available for students to grow in the garden. Another limitation of the research is that the experimental group and the control group were not students of the same grade, so there might have existed some difference in the two groups' professional knowledge accumulation and understanding of pro-environmental design, and this might have led to a certain bias in the questionnaire survey results.

**Supplementary Materials:** The following supporting information can be downloaded at: https://www.mdpi.com/article/10.3390/su14042164/s1, S1: Questionnaire and the result of Q1–Q12.

**Author Contributions:** J.X. designed the teaching methods and X.W. designed the content. They worked together to draft and revise the paper. All authors have read and agreed to the published version of the manuscript.

**Funding:** This research was funded by the Jiangsu Provincial Education Science "Thirteenth Five-Year Plan" Special Project, grant number C-a/2016/01/26.

**Institutional Review Board Statement:** The study was conducted in accordance with the Declaration of Helsinki, and approved by the Ethics Committee of Changzhou Institute of Technology (protocol code 2021-2-01, 13 February 2021).

**Informed Consent Statement:** Informed consent was obtained from all subjects involved in the study.

**Conflicts of Interest:** The authors declare no conflict of interest. The funders had no role in the design of the study; in the collection, analyses, or interpretation of data; in the writing of the manuscript or in the decision to publish the results.

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
