# Peer review of "Development of Landscape Architecture Design Students’ Pro-Environmental Awareness by Project-Based Learning"

_sustainability, doi:10.3390/su14042164_

Round 1

Reviewer 1 Report

The manuscript covers an interesting subject. However, there are issues about the methodology, theoretical underpinning, and contributions. I am particularly concerned about the novelty and rigor of the proposed study. The paper reads more like a report than a publishable paper in an academic journal.

1, I recommend changing Project-based Teaching into Project-based Education (as both students and teachers are involved in this concept), climate warming into climate change or global warming (as they are more commonly used).

2, This manuscript does not contain a well-structured review of relevant concepts necessary to understand the purpose of the study and the existing gap in knowledge.

3, a sub-section on research design is missing. Why did the authors choose this method, and how each steps were designed?

4, project-based teaching is not new. What makes the authors' case special is unclear (it was the first time that project-based pro-environment teaching methods had been used). It's unreliable to rely on students ' perceptions to validate the research solely. Interviews, focus group discussion, or other methods are needed. I am not convinced that the students developed a deeper understanding. Therefore, I am unsure if other lecturers can get inspiration from this case.

5, theoretical contribution and practical implication should be added to the abstract and 4. Discussion and conclusion. Overall, the discussion needs to be written in a manner that will respond to the research questions and highlight the contributions by reflecting the findings ON EARLIER RESEARCH. This is what a "discussion" means!

6, Professional proofread is strongly recommended. Many sentences are translated from another language and not easy to understand.

Author Response

Thank you so much for your advice. It has helped us to reflect on the paper and further improve it in both content and language. 

Reviewer 2 Report

The authors are to be acknowledged for exploring project-based learning in developing college students' pro-environment awareness during landscape architecture course instruction.

Their selection of this research topic and the experimental methodology chosen denote a well planned and performed research design. 

Their current findings tend to reflect a good starting point to extend this line of study in future research related to landscape architecture.

My only suggestion would be to kindly proceed with a few minor English language editing for their manuscript to reach absolute excellence (e.g. lines 23-24 provided them with an opportunity, lines 70-71 is and not was within the domain of, line 157 advice/instruction might be a better word to use instead of counsel, etc.). I extend my best wishes to the authors in their pursue of this research work in time ahead. 

Author Response

Thank you for your advice. It has helped us to reflect on the paper and further improve it in content and language.

Reviewer 3 Report

  • You start of the manuscript discussing project-based teaching; I would recommend using the term from the education literature -- Project-Based Learning (PBL) -- so as to not confuse a reader with an education background
  • Similarly, when you discuss Zone of Proximal Development (ZPD) you need to use the term as it is in the literature so as to not confuse the reader
  • Line 83: Where you state "recommended educational methodology", I would suggest using the term approach instead of methodology 
  • Figure 1: This is very helpful in presenting to the reader how your frameworks are aligned with the course progression
  • Lines 143-215: Here you do really lose the reader. I would highly recommend breaking up this description more for the reader; consider using more tables / figures / diagrams to explain the progression of the course and how the experimental and control groups proceeded through it -
  • The reader isn't introduced to the differences / concepts of conservation (e.g., eliminating litter) and sustainability until Table 1 outlines "environment-unfriendly behaviors in investigated parks"; earlier in the manuscript you need to distinguish -- then connect - -these concepts for the reader. I'd recommend citing relevant, recent literature on these two concepts as well. In looking at the unfriendly behaviors to the environment (Table 1) your class seems much more concerned with conservation than climate change mitigation per se as it is written currently. 
  • You must run significance testing between the experiment and control groups; frequencies are not enough here to support your claims, as stated (you may also modify your stated outcomes / impacts of the course as appropriate)
  • Figure 2: cite the photographer(s) of these images (even if it is a class instructor or student)

Author Response

(The authors gave the same response as above.)

Round 2

Reviewer 1 Report

1, A literature review on PBL and ZPD is lacking. Why this research is necessary, and what potential contribution could be made.
2, The language is still not satisfactory. For example, Pro-environmental awareness is compose of pro-environmental knowledge. Several "Pro-environmental awareness" could be replaced by "it."

Author Response

Thank you so much for your advices. 

Reviewer 3 Report

Thank you for making these revisions; there are two remaining concerns to be addressed.
(1) Significance Testing:
This is in reference to my original review calling for significance testing. Running a Cronbach Alpha indeed helps clarify any internal reliability concerns with your Likert-like instrument (thank you for adding this - it was needed :).  However, given that your paper is making claims that this structure was a success and should be considered for wider adoption, it is essential to show the significance of the difference between the control and the experimental groups here. To support your claims, you will need significance testing between the post-test control and experimental findings.  Take care to explain your testing here, as the Likert-like scale deployed here is both categorial and ordinal in nature. 

(2) Higher Education  / Design Curriculum
To strengthen your additions relating to my original review, you may want to briefly describe and site the literature that discussed how design studios classes have not grappled enough with drawing from evidence-based education practice and the complexity of design itself (thus reinforcing how innovative your class was in doing so) -- some citations that comes to mind (but there may be others that are a better fit):

Michael W. Meyer & Don Norman (2020). Changing design education for the 21st century. 

Tsungjuang Wang A New Paradigm for the Design Studio (2010)   Dreamer. Peggy (2020) Design pedagogy: the new architectural studio and its consequences. Architecture, Media, Politics & Society Proceedings (20200801). https://doi.org/10.14324/111.444.amps.2020v18i1.002    

Author Response

Thank you so much for your advices. 

Round 3

Reviewer 1 Report

The paper has been improved but can be better. The literature review should show the state of art, what has been done and what is missing. Only touching the definition and introduction of the concepts is not sufficient. The MDPI language editor should check the language carefully. Line125-128 and Line 425-427 should not be the same.

Author Response

Thank you very much.

We have improved the paper according to your advice, citing more relevant research in the introduction and changing the sentence in the discussion and conclusion that repeated a sentence in the introduction. Hope these minor revisions can make the paper better. Thanks again for your help.
